

# Human disturbance caused stronger influences on global vegetation change than climate change

Xianliang Zhang[1,2] and Xuanrui Huang[1,2]

[1] College of Forestry, Hebei Agricultrual University, Baoding, China
[2] Long-term Forest Silviculture Research Station in Saihanba, Chengde, China

## ABSTRACT

Global vegetation distribution has been influenced by human disturbance and climate change. The past vegetation changes were studied in numerous studies while few studies had addressed the relative contributions of human disturbance and climate change on vegetation change. To separate the influences of human disturbance and climate change on the vegetation changes, we compared the existing vegetation which indicates the vegetation distribution under human influences with the potential vegetation which reflects the vegetation distribution without human influences. The results showed that climate-induced vegetation changes only occurred in a few grid cells from the period 1982–1996 to the period 1997–2013. Human-induced vegetation changes occurred worldwide, except in the polar and desert regions. About 3% of total vegetation distribution was transformed by human activities from the period 1982–1996 to the period 1997–2013. Human disturbances caused stronger damage to global vegetation change than climate change. Our results indicated that the regions where vegetation experienced both human disturbance and climate change are eco-fragile regions.

## INTRODUCTION

Vegetation is the most important component of the global terrestrial ecosystem. Influenced by human disturbance and climate change, global vegetation has shifted from a semi-wild terrestrial biosphere to a mostly anthropogenic biome (*Ellis et al., 2010*). The changes in the global vegetation or land use in the past were addressed in numerous studies (e.g., *Wang, Price & Arora, 2006*; *Li et al., 2017*; *Song et al., 2018*). However, few studies evaluated the individual contributions of human disturbance and climate change on vegetation changes, which is crucial to know whether the past vegetation changes are mainly caused by human or nature.

Human disturbances and climate change are the two main factors that determine the changes in regional vegetation distribution. The land surface of the Earth has been modified by human activities (e.g., farming, construction and grazing) for centuries, and it has been significantly changed by human activities (*Foley et al., 2005*; *Ellis & Ramankutty, 2008*; *DeFries et al., 2010*). More of the land surface is being transformed as the human population continues to increase (*Goldewijk et al., 2011*; *Ellis et al., 2013*). The new geological epoch

Corresponding author
Xianliang Zhang,
zhxianliang85@gmail.com,
lxzhxl@hebau.edu.cn

has been referred to as the Anthropocene (*Lewis & Maslin, 2015*) and the period in which biomes have been severely transformed by human disturbance is referred to as an anthrome (*Ellis & Ramankutty, 2008*). With the continuous expanding human settlements, the distribution of vegetation types across the globe has changed markedly compared with the potential distribution of vegetation types which reflect vegetation distribution in the absence of anthropogenic influences.

Climate change is another factor that causes shifts in the vegetation distribution (*Kelly & Goulden, 2008*). Climate change had strong influences on the transformation of the vegetation distribution. Tropical rainforest and arctic tundra have experienced boundary changes as a result of climatic change (*Diaz & Eischeid, 2007*; *Cook & Vizy, 2008*; *Zhang & Yan, 2014a*). Widespread forest die-off from drought and heat stress increased with climate change (*Allen et al., 2010*; *Anderegg, Kane & Anderegg, 2013*). Some dying forests are likely to be replaced by other vegetation types. For instance, boreal forests are experiencing the strongest warming among forest ecosystems, and large area of boreal forest are expected to be replaced by other biomes (*Gauthier et al., 2015*). Vegetation distribution change would be severe with continuous climate warming.

The anthropogenic transformation of biomes and the terrestrial biosphere has been investigated to reflect vegetation distribution changes by comparing different biomes at century intervals (*Ellis et al., 2010*; *Ellis, 2011*), without considering the contribution of climate changes on shifting the vegetation. In addition, climate data (1700–2000) used to detect the anthropogenic transformation of biomes was almost 20 years ago (*Ellis et al., 2010*). The updated analysis is urgently needed as climate change plays an important role in shifting the vegetation.

In this study, we delimit the influence of human disturbance and climate on the distribution of vegetation based on updated data from 1982 to 2013. Accordingly, our main goals were to (1) delineate the influence of human disturbance versus climate on the vegetation distribution, and (2) identify those regions most susceptible to human disturbance in recent period.

## DATA AND METHODS

### Climate and vegetation data

Global gridded monthly mean temperature and total precipitation data were obtained from the CRU TS 4.01 dataset at $0.5° \times 0.5°$ resolution (*Harris et al., 2014*). This dataset interpolates climate data from meteorological stations distributed throughout the world to the global land area, grid-by-grid, for the period 1901–2013 and has been used in previous climate classifications (*Zhang & Yan, 2014a*). We used climate data for the period 1982–2013 as this period reflected the availability of the vegetation data.

The normalized difference vegetation index (NDVI) has been used to indicate the greenness of vegetation in numerous vegetation studies (e.g., *DeFries et al., 2000*; *Loveland et al., 2000*; *Breshears et al., 2005*; *Tucker et al., 2005*; *Zhou et al., 2014*). As a coarse measure, the differences in NDVI may mask changes to vegetation species composition; however, it

has limited influences on identifying vegetation types. It is defined as

$$NDVI = (NIR - RED)/(NIR + RED),$$

where NIR and RED are the amounts of radiation in the near-infrared and red regions, respectively. The NIR and RED reflectances should be corrected for atmospheric effects. The NDVI values range from $-1$ to 1, where negative values correspond to an absence of vegetation and positive values indicate vegetated land.

Monthly mean NDVI data at $0.0833° \times 0.0833°$ spatial resolution were retrieved from the Advanced Very High Resolution Radiometer (*Pinzon & Tucker, 2014*) based Global Inventory Modelling and Mapping Studies dataset (https://ecocast.arc.nasa.gov/data/pub/gimms/3g.v0/) for the period 1982–2013.

The high-resolution climate data was not available for the vegetation classification. Hence, the NDVI data was up-scaled to match the resolution of climate data. The NDVI data were up-scaled by calculating the arithmetic mean of the nearest neighbor grids over a six-by-six window because one grid cell of climate data ($0.5° \times 0.5°$) covers six-by-six grid cells in NDVI data ($0.0833° \times 0.0833°$).

## Separation of climate- and anthropo-driven vegetation changes

How to delimit the influence of human disturbance and climate on the vegetation was outlined in the sketch of Fig. 1. The vegetation distribution in the real world could be represented by existing vegetation types which reflected the vegetation distribution under human influences (*Zhang et al., 2017a*). Existing vegetation types include effects of human influences while potential vegetation types exclude these effects. The real vegetation changes could be identified by the changes in existing vegetation over the two periods. The potential vegetation changes are mainly caused by climate changes. The climate-driven vegetation changes could be reflected by the changes in potential vegetation over different periods. The anthropo-driven vegetation changes can be identified by the difference between the changes in potential and existing vegetation.

## Potential vegetation distribution

The potential vegetation was generally defined based on climate variables (*Köppen, 1936*; *Holdridge, 1947*; *Holdridge, 1967*; *Box, 1981*; *Box, 1996*; *Ramankutty & Foley, 1999*; *Beck et al., 2005*; *Baker et al., 2010*; *Ellis et al., 2010*; *Levavasseur et al., 2012*); therefore, the potential vegetation types could be represented by corresponding climate types. Climate types could be objectively classified to different global climate types based on the monthly attributes using the K-means clustering method (*Mahlstein, Daniel & Solomon, 2013*; *Zhang & Yan, 2014a*; *Zhang & Yan, 2014b*; *Zhang & Yan, 2016*; *Zhang, Yan & Chen, 2017b*). Monthly mean temperature and monthly total precipitation were used as input multivariables that consisted of an $n \times 24$ matrix **X1**:

$$\mathbf{X1} = \begin{bmatrix} T_{11} & \cdots & T_{1m} & P_{11} & \cdots & P_{1m} \\ \vdots & \vdots & \vdots & \vdots & \vdots & \vdots \\ T_{n1} & \cdots & T_{nm} & P_{n1} & \cdots & P_{nm} \end{bmatrix}$$

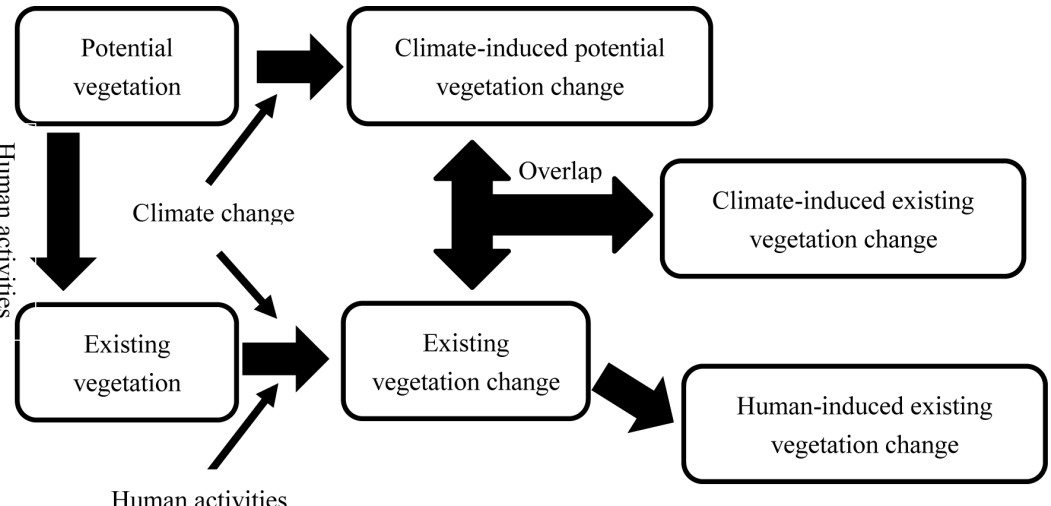

**Figure 1  Sketch map for delimiting individual contributions of human disturbances and climate changes on vegetation.**

where $T$ is monthly mean temperature, $P$ is monthly mean precipitation, $m$ is 12, and $n$ is the number of all the grid cells in the global land area, except the Antarctic. The rows in **X1** represent the monthly attributes, while the columns represent the number of grid cells. The names of vegetation types were designated by referring to the Koeppen classification (*Kottek et al., 2006*).

## Existing vegetation distribution

The existing vegetation types were classified based on climate and NDVI data using the K-means method (*Zhang et al., 2017a*). An $n \times 36$ matrix **X2** was constituted by monthly mean temperature, monthly total precipitation and monthly mean NDVI:

$$\mathbf{X2} = \begin{bmatrix} T_{11} & \cdots & T_{1m} & P_{11} & \cdots & P_{1m} & \text{NDVI}_{11} & \cdots & \text{NDVI}_{1m} \\ \vdots & \vdots & \vdots & \vdots & \vdots & \vdots & \vdots & \vdots & \vdots \\ T_{n1} & \cdots & T_{nm} & P_{n1} & \cdots & P_{nm} & \text{NDVI}_{n1} & \cdots & \text{NDVI}_{nm} \end{bmatrix}$$

where $T$ is monthly mean temperature, $P$ is monthly mean precipitation, NDVI is monthly mean NDVI, $m$ is 12, and $n$ is the number of all the grid cells in the global land area, except the Antarctic.

## Temporal changes in the influences of human disturbance and climate change on vegetation distribution

*Fraedrich, Gerstengarbe & Werner (2001)* suggested that an interval of at least 15 years is required to detect temporal changes in the geographical distribution of climate types. Thus, the period from 1982 to 2013 was split into two periods (1982–1996 and 1997–2013), and the existing and potential vegetation types were classified over the two periods to check the temporal changes in the influences of human disturbance and climate change on vegetation distribution.

The differences between the existing vegetation distribution and the potential vegetation distribution over the period 1982–1996 reflected the vegetation changes from no human influence period to the period 1982–1996. The differences between the existing vegetation over two periods 1982–1996 and 1997–2013 reflected the existing vegetation changes from the period 1982–1996 to the period 1997–2013.

The potential vegetation changes were indicated by the changes in climate types over the two periods. However, whether or not the potential vegetation change was reflected in the real vegetation changes should be verified by checking the overlapped changes in both potential vegetation and existing vegetation. When changes were detected in both the potential and existing vegetation, they were identified as the influence of climate changes on vegetation. The impacts of human disturbance on the vegetation distribution could be identified by the differences between the existing vegetation changes and the potential vegetation changes.

## RESULTS

Global existing vegetation types were defined for the period 1982–1996 (Fig. 2A) and the period 1997–2013 (Fig. 2B). Changes in the distribution of existing vegetation were found worldwide, except in the polar and desert regions, from the period 1982–1996 to the period 1997–2013 (Fig. 2C). The largest changes in vegetation types were found in central Africa, eastern China, western America and Australia. The least changes in the vegetation were found in tropical rainforest, tropical and temperate deserts, frigid deciduous coniferous forest and polar frost.

The distribution of potential vegetation over the period 1982–1996 (Fig. 3A) was similar to that over the period 1997–2013 (Fig. 3B). Changes in potential vegetation mainly occurred on the boundaries between adjacent types of vegetation from the period 1982–1996 to the period 1997–2013 (Fig. 3C). Obvious boundary changes were seen between tropical rainforests and tropical dry forests, between tropical deserts and the Sahel, and between temperate deciduous and evergreen forests. After comparison with the existing vegetation changes, actual changes in existing vegetation caused by climate variations were only detected in a limited number of grid cells (Fig. 3D). These grid cells were distributed worldwide, mainly in the ecotones.

A large area of vegetation was transformed by human disturbances (Fig. 4C) by changing the potential vegetation (Fig. 4B) to the existing vegetation (Fig. 4A) from no human influence period to the period 1982–2013. The impact of human disturbance on the vegetation distribution occurred worldwide from the period 1982–1996 to the period 1997–2013 (Fig. 5). The human-induced vegetation changes were not only seen at the boundaries of vegetation types, but also within the regions of vegetation types. About 3% of total vegetation distribution was transformed by human activities from the period 1982–1996 to the period 1997–2013. The largest changes in vegetation were found in the eastern China, central Africa and western America.

Eastern China was selected to verify our results (Fig. 6). The NDVI changed from −0.04 to 0.04 in the grid cells where the changes in existing vegetation occurred (Fig. 6B).
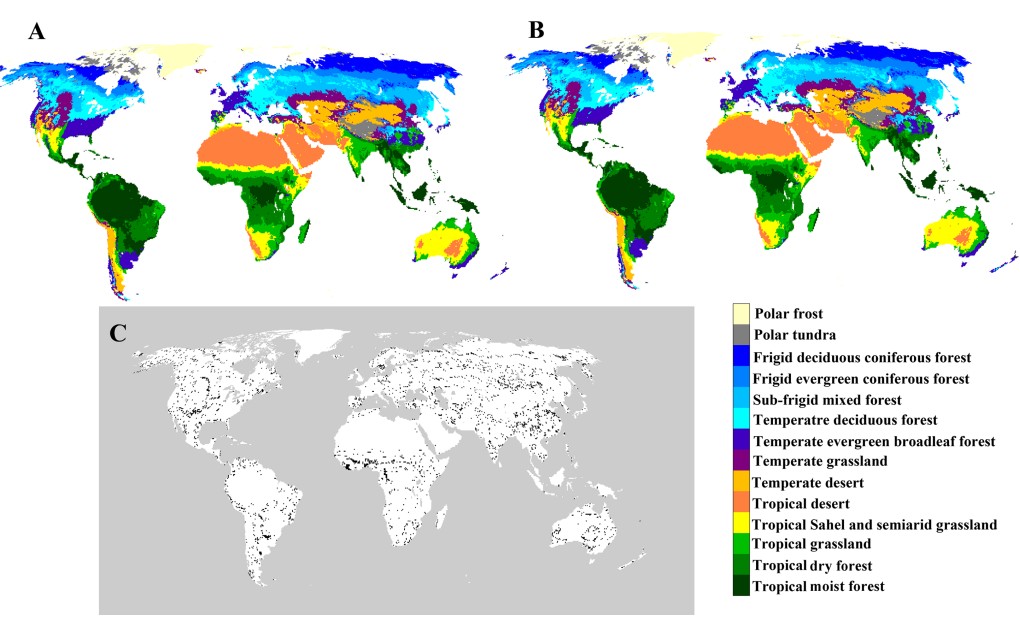

**Figure 2** **Geographical distribution of climatic vegetation types for (A) the period 1982–1996 and (B) the period 1997–2013 and (C) the differences between them.** The black regions are those that have undergone transformations in vegetation type.

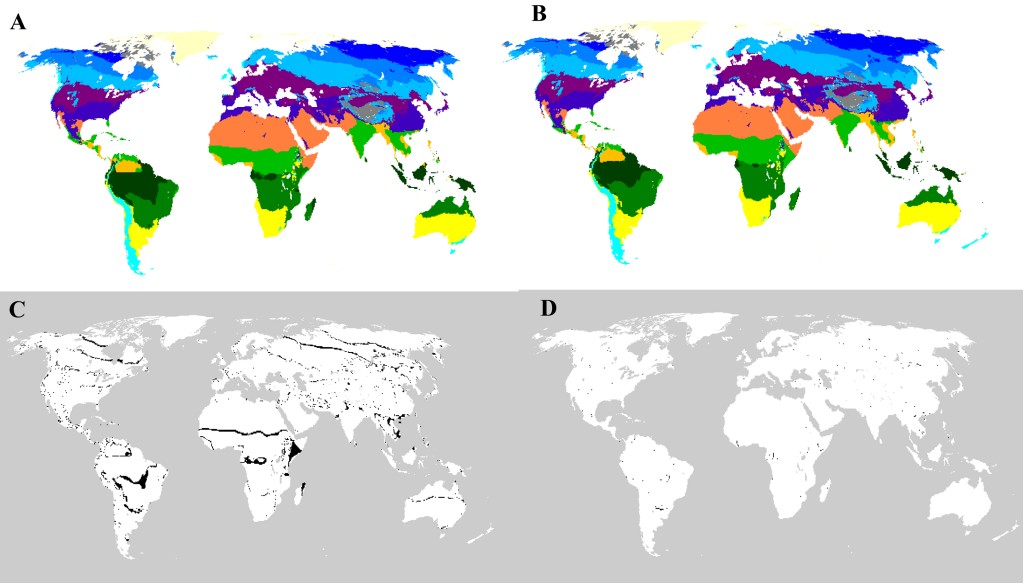

**Figure 3** **Geographical distribution of potential vegetation types for (A) the period 1982–1996 and (B) the period 1997–2013 and (C) the potential and (D) actual changes between them.** The colors used for the potential vegetation types represent the same meanings as in Fig. 2. The black regions are those that have undergone transformations in vegetation type.

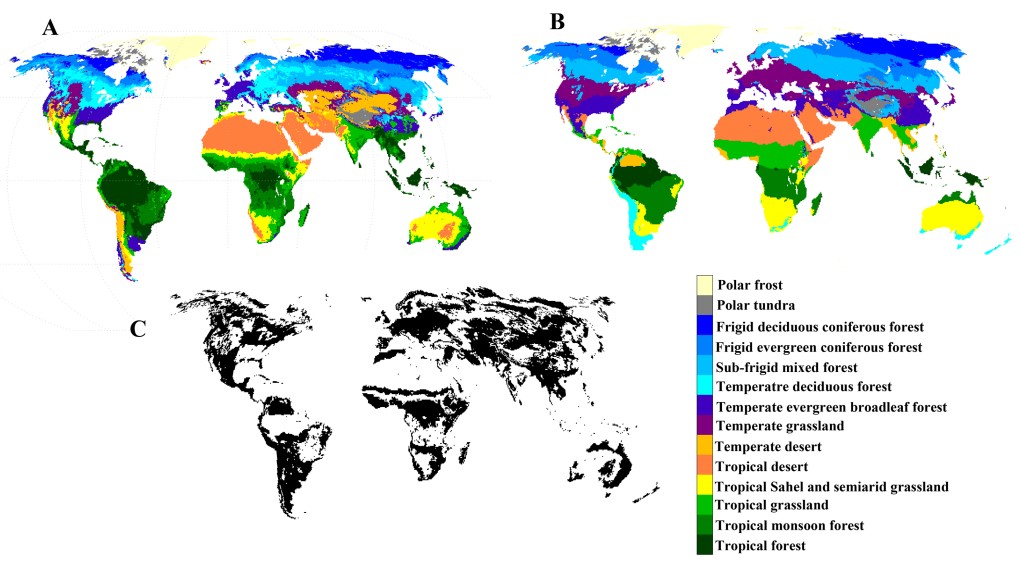

**Polar frost**
**Polar tundra**
**Frigid deciduous coniferous forest**
**Frigid evergreen coniferous forest**
**Sub-frigid mixed forest**
**Temperatre deciduous forest**
**Temperate evergreen broadleaf forest**
**Temperate grassland**
**Temperate desert**
**Tropical desert**
**Tropical Sahel and semiarid grassland**
**Tropical grassland**
**Tropical monsoon forest**
**Tropical forest**

**Figure 4 Geographical distribution of (A) existing vegetation types and (B) potential vegetation types over the period 1982–2013, and (C) the differences between them.** The differences show the effects of the human activity on the distribution of vegetation types from no human influenced period to 1982–2013.

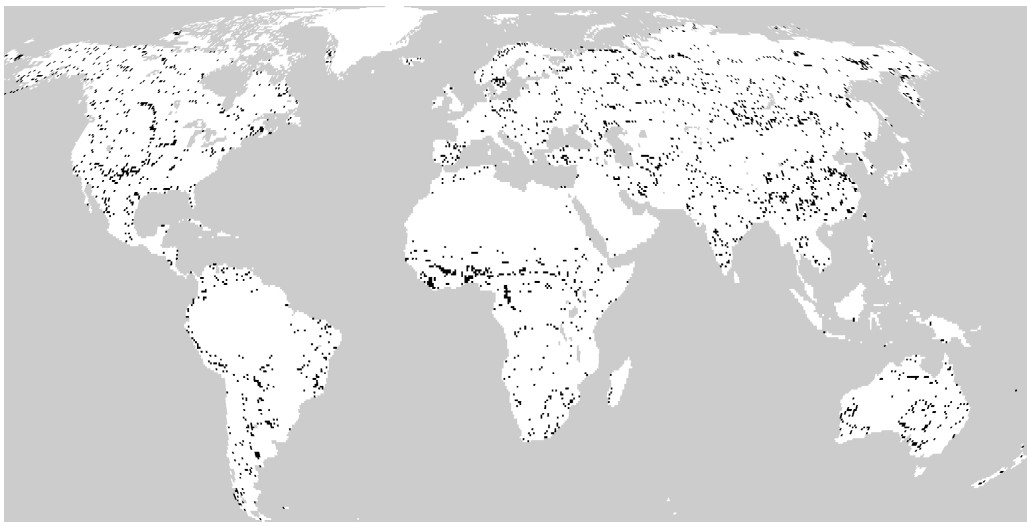

**Figure 5 Impact of human activity on the distribution of vegetation types over the period 1982–2013.** The black regions are those that have undergone transformations in vegetation type as a result of human activity.

The changes in vegetation type were verified by the changes in the NDVI (Fig. 6C). The vegetation changed in the regions where the changes in vegetation types were detected. The actual vegetation changes were compared with the land use change between 1990 and 2000 reported by *Liu et al. (2002)*, Fig. 6D. The vegetation changes detected by two studies were mainly concentrated in similar regions. The areal changes in certain vegetation types

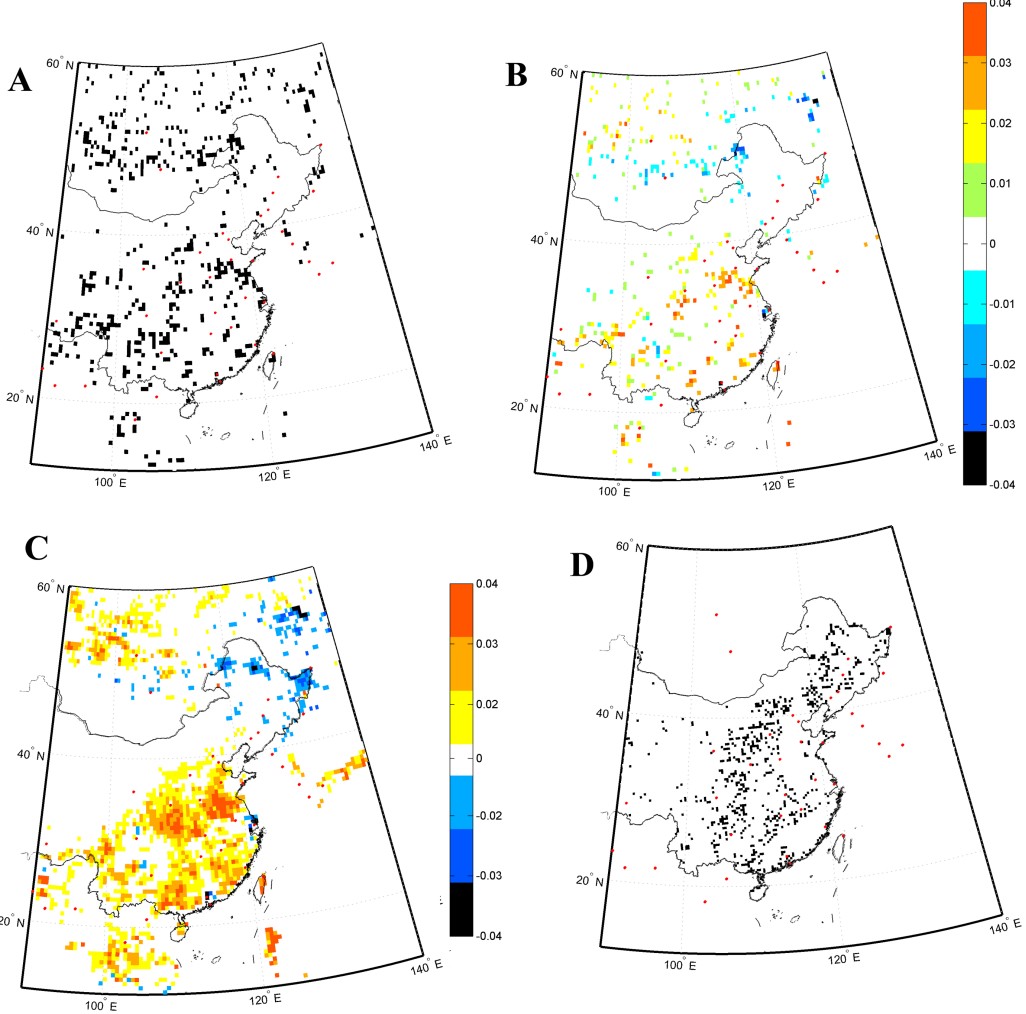

**Figure 6** (A) Changes in the existing vegetation in eastern China, (B) changes in the NDVI in regions where existing vegetation changes were detected, (C) changes in the NDVI over the whole region, and (D) changes in land use detected using the land use data of Liu. The red dot represents big city in eastern China.

were similar to those obtained using Liu's land use data (Table 1). Larger changed areas of vegetation types were detected in this study than in *Liu et al. (2002)*, because more detailed changes in land use could be detected using the land use data with higher resolution and more detailed vegetation types (24 types).

## DISCUSSION

The relative contributions of human disturbance and climate change on the vegetation changes was separated by comparing the differences between existing vegetation and potential vegetation. Potential vegetation distribution is the vegetation without human disturbances, and it was similar to potential natural vegetation in 1700 as defined by *Ramankutty & Foley (1999)* and *Ellis et al. (2010)*. The existing vegetation types were

**Table 1 Comparison of changed area of vegetation types in eastern China detected in this study to those detected using the land use data of *Liu et al. (2002)*.** Liu's land use represents the land use data of *Liu et al. (2002)*.

| | Changed area of vegetation types ($10^4$ km$^2$) | |
| --- | --- | --- |
| | This study | Liu's land use |
| Temperate grassland | 7.5 | 6.4 |
| Temperate Evergreen broadleaf forest | 7.7 | 6.1 |
| Temperate Deciduous forest | 6.7 | 4.6 |
| Sub-frigid mixed forest | 6.4 | 4.9 |
| Frigid evergreen coniferous forest | 3.3 | 2.5 |

classified based on both vegetation and climate data to reflect the connection between vegetation and climate, without using the method that defines the vegetation types based on NDVI data (*DeFries & Townshend, 1994*; *Lu et al., 2003*). Potential vegetation types and their corresponding existing vegetation types were easily compared because they were classified by the same method.

Human disturbance has influenced vegetation for several centuries. The human-induced vegetation changes from no human influenced period to the period 1982–2013 was similar to those reported human-induced land degradation (*Bai et al., 2008*). The transformations of vegetation caused by human activities mainly through farming, construction and grazing (*Barger et al., 2018*). The vegetation was mainly transformed into cropland, pasture and constructions due to human activities (*Ellis, 2011*). Transitions in land use before 1900 mainly occurred in China, India, Europe, North America and Australia (*Ellis et al., 2010*). Transformations in the distribution of vegetation accelerated when rapid growth in the human population increased the pressure to expand the amount of pasture and farmland. These regions are the key zones with respect to the anthropogenic transformation of vegetation from the no human influence period to the period 1982–2013.

The impacts of human disturbance on vegetation were seen worldwide from the period 1982–1996 to the period 1982–2013, except for the polar and desert regions. The areas that were most affected by human disturbance were those with the highest population densities, including Europe, western North America, Central Africa, southern South American, eastern Australia and eastern China. The expansion of pasture mainly took place in central Asia, Australia, southern Africa and in the tropical Sahel and subsequent overgrazing led to transformations in the vegetation cover. The amount of cropland has expanded markedly in North and South America, Europe, southern Australia, northeast China and southern Asia. Grassland has been replaced by farmland in Europe and in North and South America. Cropland has expanded into the shrublands of Australia. The main change in land use in eastern China has been the replacement of forest and grassland with cropland (*Zhang et al., 2016*; *Zhang et al., 2017c*). However, human disturbances which only influenced the vegetation structure could not be reflected by our results because no shift in vegetation types could be detected.

The boundaries of some potential vegetation types were altered by climate changes. A shifting of ecotones seems more likely the result of climate change influencing areas that are borderline based on climate, thus, a small shift in climate is likely to result in a change in ecotones rather than in central vegetation areas. The vegetation in these ecotones was under pressure and would show further changes over time. However, the influence of climate change on vegetation was limited from the period 1982–1996 to the period 1997–2013 because climate-induced vegetation shift was not viable over short periods, except when there was an abrupt climate shift. Moreover, the coarse resolution of the data restricted to detect the detail vegetation changes induced by climate change. Climate warming and hot drought would likely cause species composition of regional vegetation (*Breshears et al., 2005*; *Allen et al., 2011*), not vegetation type change (e.g., forest to grassland), which could not be visible over a short period. However, widespread increased tree mortality has been found in some forest ecosystems because of climate warming (*Adams et al., 2010*; *Van Mantgem et al., 2009*), and large area of vegetation showed greening or browning trends (*De Jong et al., 2013*). The impact of climate change on vegetation would be more visible in these regions over a longer period (e.g., 200 years).

The vegetation would return to what kind of potential forest type if there is no human disturbance can be referred by the potential vegetation types. The potential vegetation type can restore itself over a period with either limited or no human interference. For instance, abandoned farmland in northeast China can transform back to forest cover. These transformations caused fundamental vegetation changes, and could be reflected in our results. There is a large potential in global tree restoration if human disturbance was limited, which is consistent with the results reported by *Bastin et al. (2019)*.

The global existing vegetation was seriously transformed from the period 1982–1996 to the period 1997–2013. About 3% of global vegetation have changed their types in the past 30 years. Human disturbance caused stronger influences on vegetation changes than climate change, which is consistent with a study in northern forests (*Danneyrolles et al., 2019*). However, the influences of climate changes on vegetation distribution could not be ignored. The regions that were influenced by both human disturbance and climate change are vulnerable to vegetation changes in the future.

## CONCLUSION

The effects of human disturbance and climatic change on the distribution of global vegetation types could be separated using the proposed method. A large area of vegetation was transformed by human disturbances from no human influenced period to the period 1982–2013. About 3% of total vegetation distribution was transformed by human activities from the period 1982–1996 to the period 1997–2013. However, the influence of climate change on vegetation was limited from the period 1982–1996 to the period 1997–2013. Therefore, human disturbances caused stronger damage to global vegetation change than climate change.

### Funding

This work was funded by the National Key Research and Development Program of China (2017YFD060040301), the National Natural Science Foundation of China (grant numbers 41601045, 31570632, 41571094) and the talents introduction program in Hebei Agricultural University (YJ201918). The funders had no role in study design, data collection and analysis, decision to publish, or preparation of the manuscript.

### Grant Disclosures

The following grant information was disclosed by the authors:
National Key Research and Development Program of China: 2017YFD060040301.
National Natural Science Foundation of China: 41601045, 31570632, 41571094.
Hebei Agricultural University: YJ201918.

### Competing Interests

The authors declare there are no competing interests.

### Author Contributions

- Xianliang Zhang conceived and designed the experiments, performed the experiments, analyzed the data, contributed reagents/materials/analysis tools, prepared figures and/or tables, authored or reviewed drafts of the paper, approved the final draft.
- Xuanrui Huang contributed reagents/materials/analysis tools, authored or reviewed drafts of the paper.

### Data Availability

The following information was supplied regarding data availability: The data used in this study are public data. The NDVI data is available at: https://ecocast.arc.nasa.gov/data/pub/gimms/3g.v0/. The CRU climate data is available at: https://crudata.uea.ac.uk/cru/data/hrg/.

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
