# Peer review of "Human disturbance caused stronger influences on global vegetation change than climate change"

_PeerJ, doi:10.7717/peerj.7763_

## Round 0.1 · original submission · Major Revisions

Your manuscript entitled "Global vegetation change is more affected by direct human activity than by climate change" has now been seen by 3 referees, whose comments are attached. The referees acknowledge the potential interest of your work, but between them, they also raise a number of concerns, which must prevent us from offering to publish the paper in its present form.

The referees’ reports seem to be quite clear. Naturally, we will need you to address all of the points raised.

The writing is not clear enough, and there are some grammatical and structural mistakes in the manuscript.

·

Basic reporting

In general, the paper is well written and organized. The English has few grammatical errors and I have provided comments on needed corrections directly in the manuscript.
The authors present argument as to why the study is important set up the reasoning behind the experimental design. While the authors do provide some references regarding changes brought about by “human activities”, I think the discussion could be improved by reviewing the Key Findings of the IPBES assessment on land degradation and restoration (IPBES (2018): The IPBES assessment report on and degradation and restoration. Montanarella, L., Scholes, R., and Brainich, A. (eds.). Secretariat of the Intergovernmental Science-Policy Platform on Biodiversity and Ecosystem Services, Bonn, Germany. 744 pages.

Experimental design

Lines 61-63. The goals are clearly stated and general approach is clearly described.

Lines 72-79, the authors should acknowledge that differences NDVI reflect differences in “greenness”, but as a coarse measure, this may mask changes to vegetation species composition

Lines 93-96. This sentence appears to be contradictory, or at the least confusing, in relation to the design premise. While it is clear that changes to “potential” vegetation could reflect influences of climate change, stating that “climatic vegetation types include effects of human influences” is confusing. I would suggest changing terminology from “climatic vegetation” to “existing vegetation”, as that is what I think is meant.

Line 118. Again, the authors state that climate and NDVI data were used to classify “climatic vegetation types”, yet, earlier they state that climatic vegetation reflects human influences.

Validity of the findings

Lines 147-152, especially lines 149-150. A shifting of ecotones seems more likely the result of climate change influencing areas that are, literally, borderline based on climate...thus, a small shift in climate is likely to result in a change in vegetation in ecotones rather than in central veg areas. Thus, contrary to the idea that changes due to climate change could not be detected in the short time frame, what the authors report is, in fact, observation of changes resulting from climate change. The authors statement of requiring a longer time period should be applied only to the central portion of the biome.

Lines 159-160. This sentence employs circular reasoning…changes in NDVI is how changes in vegetation were defined.

Lines 168-169. I am not convinced that the authors did, indeed, completely separate human activity effects from climate change; I think the authors presented data that is consistent with hypothesis that changes at borders of biomes is actually due to climate change.

Lines 196-201. Contrary to what the authors argue in these lines, I think they have in fact documented changes in vegetation due to climate change. While their general conclusion that human activities (primarily land use change) is a larger factor in vegetation changes is demonstrated, I think it would make for a more nuanced discussion if the authors embrace the fact that they have demonstrated vegetation changes due to climate change.

Additional comments

I find this a very interesting study and I would recommend it for publication, provided that the authors better acknowledge that they have documented changes due to climate change.

Reviewer 2 ·

Basic reporting

The article aims to separate the influences of human activity and climate changes on the vegetation changes. This is an interesting topic. The writing is not clear enough. In addition, there are some grammatical and structural mistakes in the manuscript and I am not able to highlight/revise all of them. So, please revise the manuscript and improve it considerably in terms of grammatical and structural mistakes. There are important assumptions here about whether vegetation changes are affected by human activity and climate changes— and they should always be stated. There are few empirical studies and many do not confirm that assumption.
The literature is suitable for the purpose of the article.

Experimental design

1) Within Scope of the PeerJ journal.
2) Research question well defined, relevant & meaningful. It is stated how the research fills an identified knowledge gap.
3) Structure conforms to technical & ethical standard.
4) Although the experimentation is appropriate, the paper would gain in interest if it would provide a more in-depth discussion about the selected topic.

Validity of the findings

1) Data is robust, statistically sound, & controlled.
2) Conclusions are well stated, linked to original research question & limited to supporting results.
3) Speculation is welcome.

Additional comments

1. The Introduction & Data and methods need to be re-organized and written in a clear way. These severely limits any reader to understand what (and how) this study was made.
2. A sketch map of this study should be added in the revision.
3. The authors should thoroughly address the novelty and importance because so many works have focused on this point.
4. The abstract should be redone, it is hard to follow at present.
5. The section of introduction did not well review and cite existed works.
6. In fact, the author should examine the applicability and generality of normalized difference vegetation index (NDVI). Sometimes, the specific thresholds are necessary. Some other vegetation indices (EVI, DVI…) may be useful.
7. The authors conducted the resampling processing to all input ecogeographical factors (0.5°×0.5°). Why the finer resolution (0.0833°×0.0833°) was not considered in this study? In addition, the author should illustrate why the window size of 6×6 (which method) was used.
8. It is true that the vegetation changes are influences by both human activities and climate changes in the real world. The author used the specific method which mentioned in your study is not clear enough. It is noted that this section is critical for this study. I think it should be described in detail.
9. Why CRU TS 3.40 dataset were applied in your study specifically? There are some other datasets available.
10. The matrix X should be listed in order.
11. The results and discussions are weak and unsatisfied, the author should make the best effort to modify these sections. I want to see a considerable improvement in this regard.
12. The section of conclusion is missing, please address this problem.
13. The conclusion I not well stated and is written more as an abstract. A clear conclusion needs to be written.
14. The section of validation is not well described.
15. All tables/figures: Caption needs to completely explain the table, including what the abbreviations mean. Work on text clarification and formatting, improve figures, list of the abbreviations would be useful in this case to easily understand the text.

Reviewer 3 ·

Basic reporting

Line 32, it is a little odd to see "human activity" and "climate change" as two separate things because we usually assume the changing climate is mainly due to human activities. Probably authors can use "perturbation", "land use change" or "human disturbance" instead of a general "human activity".

Line 33, maybe use "construction" instead of "building", and in line 178 too

Line 34, the two parts of this sentences seem to be the same. "modified" meaning "changing" right?

Figure 2. Did authors see a movement of boundary lines? instead of reporting changes.

Figure 3. Did not authors separate the study period to two? how did authors calculate the impacts from 1982-2013? was that mentioned in the Method section?

Figure 4, it might be too much, but maybe can replace the blank background map of China to contour background or label some big cities to show near which location has the large changes.

Experimental design

Line 111, so this equation does not have parameters that account for interannual or long-term variation changes but only seasonal changes? and the monthly mean temperature or precipitation is the average of 15 years period?

Line 128-129, if an interval of at least 15 years is needed, why having more years that less than 15 years for those two periods? why not use 1982-1990 vs. 2006-2013 so those two periods have at least 15 years interval?

Validity of the findings

Line 144, how much is the largest change? I compared the Figure 1a and 1b, very similar. maybe because of the large scale. The 1c graph of difference is very helpful but I suggested to add the percent change scale on it or even color it to show difference levels of changes.

Line 186, the starting data of this study is in 1982, so this sentence "Transformations in the distribution of vegetation accelerated after the industrial revolution..." can not be written or proved in this study. The Industrial Revolution took place from the 18th to 19th centuries.

Line 208, yes but no. human activity already disturb the environments for the growth of potential vegetation type. If we return the farmland to forest, it might be a different forest type comparing with the potential vegetation type. For example, under the warmed environment already, we can only return abandoned lands to deciduous forest but not conifers.

Line 212, how to define this "low intensity of human interference"?

Adding a Conclusion section would be better, and is common for articles published in PeerJ.

Additional comments

This study reports and compares the changes in vegetation types due to changing climate only vs. land use change only. It concludes that land cover change causes more severer damage to global vegetation change than climate change. I find this paper has important values to the society.

---

## Round 0.2 · accepted · Accept

You follow the suggestions of the reviewers, greatly improving the article. The reviewers and I believe it is ready to be published.

·

Basic reporting

No comment.

Experimental design

With clarification of some of the terminology, as I suggested in my original review, I feel that the experimental design appears stronger than in original manuscript.

Validity of the findings

Again, I originally questioned the validity of some of the conclusions, but editing of the manuscript and clarification of methods, I find the conclusions much more convincing.

Additional comments

The authors did an excellent job of responding to my original comments. I have just a few, non-critical comments that I have edited into the attached pdf.

Reviewer 3 ·

Basic reporting

None

Experimental design

None

Validity of the findings

None

Additional comments

I find this paper can be accepted as it is.